# A Newly Incompatibility F Replicon Allele (FIB81) in Extensively Drug-Resistant *Escherichia coli* Isolated from Diseased Broilers

**DOI:** 10.3390/ijms25158347

**Published:** 2024-07-30

**Authors:** Ahmed M. Ammar, Norhan K. Abd El-Aziz, Mohamed G. Aggour, Adel A. M. Ahmad, Adel Abdelkhalek, Florin Muselin, Laura Smuleac, Raul Pascalau, Fatma A. Attia

**Affiliations:** 1Department of Microbiology, Faculty of Veterinary Medicine, Zagazig University, Zagazig 44511, Egypt; aaahmmad@vet.zu.edu.eg (A.M.A.); adel.575757@yahoo.com (A.A.M.A.); 2Animal Health Research Institute, Doki, Giza 12618, Egypt; galalpasha@yahoo.com; 3Food Safety, Hygiene and Technology Department, Faculty of Veterinary Medicine, Badr University in Cairo (BUC), Badr 11829, Egypt; adel.abdelkhalek@buc.edu.eg; 4Department of Toxicology, Faculty of Veterinary Medicine, University of Life Sciences “King Michael I” from Timisoara, 300645 Timisoara, Romania; florinmuselin@usvt.ro; 5Department of Sustainable Development and Environmental Engineering, Faculty of Agriculture, University of Life Sciences “King Mihai I” from Timisoara, 300645 Timisoara, Romania; laurasmuleac@usvt.ro; 6Department of Agricultural Technologies, Faculty of Agriculture, University of Life Sciences “ King Mihai I” from Timisoara, 300645 Timisoara, Romania; 7Animal Health Research Institute, Zagazig 44516, Egypt; fatmaadelatiamohamed@gmail.com

**Keywords:** *Enterobacterales*, extensive drug resistance, IncF plasmids, FIB-replicon allele

## Abstract

Multiple drug resistance (MDR) has gained pronounced attention among *Enterobacterales*. The transfer of multiple antimicrobial resistance genes, frequently carried on conjugative incompatibility F (IncF) plasmids and facilitating interspecies resistance transmission, has been linked to *Salmonella* spp. and *E. coli* in broilers. In Egypt, the growing resistance is exacerbated by the limited clinical efficacy of many antimicrobials. In this study, IncF groups were screened and characterized in drug-resistant *Salmonella* spp. and *E. coli* isolated from broilers. The antimicrobial resistance profile, PCR-based replicon typing of bacterial isolates pre- and post-plasmid curing, and IncF replicon allele sequence typing were investigated. Five isolates of *E. coli* (5/31; 16.13%) and *Salmonella* spp. (5/36; 13.89%) were pan-susceptible to the examined antimicrobial agents, and 85.07% of tested isolates were MDR and extensively drug-resistant (XDR). Twelve MDR and XDR *E. coli* and *Salmonella* spp. isolates were examined for the existence of IncF replicons (FII, FIA, and FIB). They shared resistance to ampicillin, ampicillin/sulbactam, amoxicillin/clavulanate, doxycycline, cefotaxime, and colistin. All isolates carried from one to two IncF replicons. The FII-FIA-FIB+ and FII-FIA+FIB- were the predominant replicon patterns. FIB was the most frequently detected replicon after plasmid curing. Three XDR *E. coli* isolates that were resistant to 12–14 antimicrobials carried a newly FIB replicon allele with four nucleotide substitutions: C99→A, G112→T, C113→T, and G114→A. These findings suggest that broilers are a significant reservoir of IncF replicons with highly divergent IncF-FIB plasmid incompatibility groups circulating among XDR *Enterobacterales.* Supporting these data with additional comprehensive epidemiological studies involving replicons other than the IncF can provide insights for implementing efficient policies to prevent the spreading of new replicons to humans.

## 1. Introduction

Antibiotic-resistant *Enterobacterales*, especially *E. coli* and *Salmonella* spp., are considered major enteric pathogens, posing a threat to global public health. They can occupy multiple niches, including human and animal hosts [1,2]. The United States Center for Disease Control and Prevention (CDC) has designated foodborne drug-resistant *Salmonella* spp. as a significant public health hazard.

Infective multidrug-resistant (MDR) *Enterobacterales* in the poultry industry have been linked to increased morbidity and mortality. The transmission of these pathogens via food may be responsible for causing foodborne illnesses and deaths.

Plasmids serve as reservoirs for multiple antibiotic-resistance genes among closely related bacterial species. Conjugative incompatibility F (IncF) plasmids have received much attention among 27 incompatibility (Inc) groups previously recognized in *Enterobacterales* alone. Plasmid incompatibility refers to multiple plasmids within one cell having the same replicon or partitioning system [3]. IncF plasmids were sub-grouped into replicons FII, FIA and FIB, which are the most frequently encountered among *Enterobacterales*, especially *E. coli*. They represent 23.1% of plasmids [4] that have been deposited in the plasmid multilocus sequence typing (pMLST) database (http://pubmlst.org/plasmid, accessed on 20 December 2023). Conjugative plasmids (CPs) transmitting antibiotic resistance genes are the major direct cause of outbreaks of antibiotic-resistant *Enterobacterales*. Actually, Gram-negative bacilli have a remarkable capacity to transfer conjugative antibiotic resistance genes [5]. Transfer of a single plasmid to a bacterium allows it to be resistant to multiple antibiotics. Therefore, plasmids harboring antimicrobial resistance genes may result in plasmid endemics [6,7]. Despite the relatively low abundance of conjugative plasmids, many resistance genes of sulfonamide (*sulI* and *sulII*), chloramphenicol (*cat*), trimethoprim (*dfr*), colistin (*mcr*), New Delhi metallo-beta-lactamase (*bla*_NDM_) and extended-spectrum β-lactamase (ESBL; *bla*_TEM_, *bla*_CTX-M_ and *bla*_SHV_) were significantly located in these plasmids [8,9,10].

The widespread nature of antimicrobial resistance (AMR) genes mediates plasmid epidemics among different members of *Enterobacterales* [11]. The bacterial-resistant phenotypes observed during epidemics can persist for months even in the absence of ongoing specific selection with risk of prolonged carriage [12]. Plasmid epidemics arise from the massive exchange of AMR genes (more than 86%) not only across other distant plasmid categories but also between bacterial chromosomes [13]. The mobility of CPs is a strong factor facilitating the accumulation of AMR genes as these plasmids possess the ability to pass across phylogenetically distant bacteria [14,15]. Transfer and stabilization of mutations in insertion sequence (IS)-associated AMR genes are affected by the use of antibiotics, which also promote bacterial conjugation [16,17,18], support bacterial fitness and persistence in adverse environmental conditions, providing virulence (bacteriocins, siderophores, cytotoxins, and adhesion factors) and antimicrobial resistance determinants [19]. Antibiotic selective pressure on commensals and pathogenic AMR *Salmonella* spp. and *E. coli* leads to clonal expansion of these strains [20]. Using oral antibiotics during the broiler production period for therapeutic, prophylactic, or subtherapeutic growth promotion purposes modifies gut microbiota [21]. Lateral transfer of CPs to recipient cells confers genes encoding antimicrobial resistance, exerts strong selective pressure, and facilitates the spread of genetic elements of insertion sequences (*IS26* and *IS1294*), integrons, and transposons [22,23].

In Egypt, IncF grouping studies have focused only on the association of resistance to an antibiotic with a specific replicon using whole-genome sequencing. However, there is a need to investigate the molecular subtyping of conjugative IncF plasmids [24]. In the present work, PCR-based replicon grouping was applied using conserved IncF primers following the GenBank entries, and the replicon type was subjected to allele subtyping [25].

## 2. Results

### 2.1. Identification of Bacterial Isolates

A total of 67 (82.72%) bacterial isolates were recovered from 81 examined liver specimens of broiler chickens, from which 31 *E. coli* (46.2%) and 36 *Salmonella* spp. isolates (53.7%) were identified. The selected drug-resistant isolates (n = 12) were identified as biotypes (Table 1) as the following: *Salmonella enterica* serovar Typhimurium (*S.* Typhimurium; n = 1), *Salmonella enterica* serovar Enteritidis (*S.* Enteritidis; n = 4), *Salmonella enterica* serovar Pullorum (*S.* Pullorum; n = 1), *Salmonella enterica* serovar Gallinarum (*S.* Gallinarium; n = 2), *Salmonella enterica* serovar Arizona (*S.* Arizona; n = 1) and *E. coli* (O125; n = 3).

### 2.2. Antimicrobial Susceptibility Patterns of Bacterial Isolates

As depicted in Table 1, *E. coli* isolates showed high resistance levels to ampicillin and colistin (83.87%), followed by ampicillin/sulbactam (80.65%) and amoxicillin/clavulanate (64.52%). However, *Salmonella* spp. isolates were highly resistant to ampicillin, colistin and ampicillin/sulbactam (86.11% each). Five isolates of *E. coli* (5/31; 16.13%) and *Salmonella* spp. (5/36; 13.89%) were pan-susceptible to the examined antimicrobial agents, and 85.07% of the tested isolates were MDR and XDR (Appendix A). Statistical analysis revealed significant variations in the resistance levels of *E. coli* and *Salmonella* spp. isolates to ciprofloxacin, azithromycin, and fosfomycin at a *p*-value < 0.05 maximizing for *E. coli* isolates (Table 1).

Twelve drug-resistant *E. coli* and *Salmonella* spp. isolates with MAR indices ranging from 0.64 to 1 were selected for IncF replicon typing (Appendix A). They exhibited shared resistance to ampicillin, ampicillin/sulbactam, amoxicillin/clavulanate, doxycycline, cefotaxime, and colistin (Table 2). Interestingly, two out of twelve (16.6%) isolates were resistant to all tested antimicrobials. On the other hand, fosfomycin was the most effective among the examined antimicrobials (58.33%). However, fosfomycin-sensitive isolates were resistant to nine or more of the examined antimicrobials. The bacterial isolates were subjected to SDS curing to detect the eliminated antibiotic resistance makers.

### 2.3. Antimicrobial Susceptibility Patterns of the SDS-Cured Isolates

The SDS-cured isolates were subjected to antimicrobial susceptibility testing to detect the cured antibiotic resistance markers, followed by IncF replicon typing (Table 3). Nine phenotypic variants were recorded after SDS treatment with a curing efficacy of 75% (9/12). Colistin, gentamycin, cefoperazone, ceftriaxone, ciprofloxacin, azithromycin, sulphamethoxazole/trimethoprim, and fosfomycin were the cured antimicrobial resistance markers among the examined antimicrobials. Gentamycin, chloramphenicol, and ceftriaxone were frequently cured.

### 2.4. Occurrence of IncF Replicons among Bacterial Isolates

The occurrence and distribution of IncF replicons among the examined isolates were investigated before and after SDS curing using PCR-based replicon typing (Table 3). All isolates carried at least one to two IncF replicons. Most isolates contained both FIA and FIB replicons. Before curing, PCR replicon typing demonstrated that FIB, FIA, and FII presented in 8/12 (66.6%), 6/12 (50.0%), and 3/12 (25.0%) bacterial isolates, respectively. Five different IncF replicon patterns were recorded; FII-FIA-FIB + and FII-FIA + FIB- were the predominant ones. After curing, 6/12 (50.0%) of the bacterial isolates showed no cured IncF replicons. The isolates with cured replicons exhibited that FIB was the most frequently detected replicon after plasmid curing. Three strains showed no plasmid curing of the Inc FIB (*E. coli* O125) and Inc FIA (*S.* Typhimurium, *S.* Enteritidis, and *S.* Gallinarium) when attributing the loss of replicons to the loss of plasmids. Two *E. coli* isolates showed no cured antibiotic resistance markers and carried the same replicon subtype (FIB81).

### 2.5. IncF (FIB) Alleles Subtyping

FIB amplicon sequences of the three *E. coli* isolates (O125) were analyzed by submitting them to the pMLST website, www.pubmlst.org/plasmid/ accessed on 20 March 2024. The new IncF replicon allele (FIB81) identified in this study was deposited in GenBank and released under an accession number OR453940. Using BLASTn, the query sequence from the plasmids displayed 98.93% identity with the reference sequence (FIB1). The four nucleotides, C, G, C, and G, in the FIB1 allele at positions 99,112,113, and 114 were substituted with A, T, T, and A, respectively, in the query sequence (Appendix A). These point mutations referred to a new FIB replicon which was identified as the FIB81 allele. The new allele (FIB81) detected in *E. coli* O125 isolated from a broiler chicken is closely related to a newly discovered FIB51 allele deposited in the pMLST database and isolated from a human case in France (Figure 1).

## 3. Discussion

The broiler industry is a great contributor to meat contamination with *Salmonella* spp. and *E. coli* as they are widely disseminated and transmitted to humans causing major health problems worldwide, particularly in developing countries [26,27]. The dissemination of antibiotic-resistance genes by *Enterobacterales* on conjugative plasmids (CPs) poses a significant threat to effective treatment. The extensive use of antibiotics accompanied with the lateral transfer of antibiotic resistance markers mediated by CPs in *Salmonella* spp. and *E. coli* has raised concern for public health. Continued usage of antibiotics and stable plasmid inheritance (plasmid stability) enhance the long-term persistence of CPs in microbial communities. The acquisition of these plasmids is often accompanied by rapid adaptation to antibiotics [28,29,30]. CPs are transferred between different genera of *Enterobacterales*, disseminating various AMR genes and acquiring different mobile genetic elements (MGEs) from chromosomes and/or other plasmids leading to the microbial evolution of resistance to antimicrobial agents [13]. The host cell harboring transferred CPs would maintain high frequencies in the populations as a consequence of antibiotic use. The selected population rapidly achieves high frequencies after the establishment of CPs packaged with resistance markers. Such a population does not respond to treatment resulting in environmental pollution with antibiotic residues transferred via processed chicken meat to humans [31].

Currently, in Egypt, the clinical effectiveness of most antimicrobials in broiler farms is limited, necessitating the use of two or more combined antimicrobials to combat infections. Limited molecular studies have been conducted concerning IncF replicons in relation to the antibiotic resistance markers among *Enterobacterales* infections in broilers. Therefore, the main aim of this study was to screen and characterize IncF replicons (FII, FIA, and FIB) in relation to antimicrobial resistance markers. This study investigated the antimicrobial resistance patterns of isolated *Salmonella* spp. and *E. coli*, as well as identifying and subtyping IncF amplicons. The results revealed that the examined drug-resistant strains carried multiple IncF amplicons with the occurrence of a novel FIB amplicon allele. To our knowledge, this is the first report in Egypt that has screened and characterized a newly IncF FIB allele.

Resistance rates to antimicrobials in Egypt have been rising over the last five years. Most strains obtained in the present study were XDR (susceptible to only one or two antimicrobial groups). They showed shared resistance to ampicillin, ampicillin/sulbactam, amoxicillin/clavulanate, colistin, doxycycline, and cefotaxime. This resistance rate is considerably higher than that of the previously reported strains, which showed high sensitivity to norfloxacin (100%), gentamycin (96.77%), and neomycin (83.87%) in *Salmonellae* isolated from retail chicken samples collected in 2017 [32].

In the present work, all strains were found to carry one to two IncF replicons. A previous study in Egypt showed that IncFIB(K) and IncFII(K) replicons were predominant in all *K. pneumoniae* strains [33]. Plasmids containing these replicons were associated with the transfer of carbapenem resistance [24]. The FIB replicon was the most predominant among the current examined strains. Previously, *E. coli* from raw milk and clinical mastitis showed plasmids containing multiple IncF replicons including IncFIA, IncFIB and IncFII; these plasmids were found to carry the FIB16 replicon-type allele (AP001918.1) [34]. The same replicon was identified in *E. coli* strains isolated from patients with respiratory infection [35]. In Nigeria, *E. coli* isolated from humans and the food chain showed IncF multi-replicon plasmids, especially those with low copy number plasmids of the FIB replicon, which were the most prevalent among the examined types (B/O, FIB, FIC, I1, K, W, X and Y) [36]. The FIB replicon followed IncFII as a major divergent replicon recognized in *Enterobacterales*, where it comprised eighty known alleles assigned in https://pubmlst.org/plasmid/ accessed on 24 May 2024. The IncFIB plasmid can widely transfer to a recipient stain; the latter persists in poultry feed chains and successfully colonizes and infects humans [37]. Herein, based on nucleotide point mutations, a novel FIB amplicon (FIB81) was identified. Implementing efficient treatment policies and infection control programs is required to prevent the dissemination of these new replicons to both humans and food chains.

The current antimicrobial resistance in Egypt may be attributed to several factors regarding the inappropriate use of antimicrobials including the following: (i) low level of adherence to appropriate antimicrobial usage, especially during the COVID-19 era [38] and lack of expertise and awareness of healthcare providers for real antimicrobial use [39]; (ii) progression of antimicrobial resistance to most drugs [34,40,41] has led prescribers to depend on the extensive use of enrofloxacin, extended-spectrum cephalosporins (ESCs), lincospectin, ceftifure, colistin, and fosfomycin as primary drugs against *E. coli* infections in diseased broilers, resembling the common use of third-generation cephalosporins in intensive care unit patients [42]; (iii) the administration of short courses of antibiotic treatment, which enhances bacterial cost effects and genetic diversity of gut commensal bacteria.

This situation might lead to the following outcomes: (i) difficulty simultaneously removing all selective pressure; (ii) after the reappearance of selective pressure, the hidden bacteria carrying CPs may re-emerge; therefore, plasmid-mediated resistance is unlikely to disappear completely from the population [43].

The following measures should be included in the control program to effectively stop the spread of IncF replicons: (i) implementing an antibiotic policy that provides veterinary prescribers with a range of antibiotic agents customized to their needs and which should be guided by laboratory susceptibility tests; (ii) monitoring the epidemiology and characteristics of resistant CPs to develop novel, effective treatment regimens, and infection control strategies [44]; (iii) taking into account the combined antibiotic bactericidal therapy of diseased broilers during the farming period, such as gentamycin/ciprofloxacin, fosfomycin/colistin, azithromycin/fosfomycin, and azithromycin/colistin [45]; (iv) inhibiting bacterial conjugation and plasmid elimination by combining antibiotic therapy with plant extracts [46,47].

## 4. Materials and Methods

### 4.1. Sample Collection

Eighty-one freshly dead birds were obtained from broiler breeding farms in different localities within the Sharkia governorate, Egypt. Liver samples (n = 81, one sample representing each bird) were aseptically collected during the period from January to April 2023, labeled, and transported to the diagnostic bacteriology laboratory at Faculty of Veterinary Medicine, Zagazig University, Egypt. The collection of samples complied with the general guidelines of Faculty of Veterinary Medicine, Zagazig University, Egypt. Written and signed consent was obtained from the owners of poultry farms to be involved in the follow-up study and those from whom chickens samples were taken to be included in the study.

### 4.2. Bacterial Isolation and Identification

Liver specimens were subjected to isolation and identification of *Enterobacterales* [48]. For isolation of *Salmonella* spp., a loopful from each specimen was inoculated into buffered peptone water (BPW; Oxoid, Cambridge, UK) and then incubated at 37 °C for 16 h. A total of 100 µL of pre-enrichment BPW culture was transferred into 10.0 mL Rappaport Vassiliadis soy broth (RV; Oxoid, Cambridge, UK) and then incubated at 42 °C for 24 h. Selective plating was applied onto xylose-lysine-deoxycholate (XLD; Oxoid, Cambridge, UK) agar and was then incubated at 37 °C for 24 h. For isolation of *E. coli*, a loopful from each liver specimen was inoculated into BPW and incubated aerobically at 37 °C for 12 h. A loopful of broth culture was streaked onto an eosin methylene blue (EMB; Oxoid, Cambridge, UK) agar plate and then incubated at 37 °C for 24 h. Developed colonies with green metallic sheen were picked up and streaked onto MacConkey’s agar plates and then incubated at 37 °C for 24 h. Fresh suspected colonial suspensions of *E. coli* and *Salmonella* spp. were biochemically confirmed using the API20E system (bioMérieux, Lyon, France). The O antigens in *E. coli* suspensions were biotyped and then subjected to serotyping using a set of O antisera at the Animal Health Research Institute, Serology Unit, Doki, Giza, Egypt as previously described [49]. Bacterial strains were preserved at −80 °C in brain heart infusion (BHI; Oxoid, Cambridge, UK) with 10% glycerol [50].

### 4.3. Antimicrobial Susceptibilities of the Identified Strains

Antimicrobial resistance markers of *Salmonella* spp. and *E. coli* were determined using the agar disc diffusion method, according to the standards and interpretive criteria described by the Clinical and Laboratory Standards Institute [51]. The antimicrobial susceptibility was examined against 13 antimicrobial discs (Oxoid, Cambridge, UK): ampicillin (AMP; 10 µg), ampicillin/sulbactam (SAM; 20 µg), doxycycline (DO; 5 µg), cefotaxime (CTX; 30 µg), chloramphenicol (C; 30 µg), azithromycin (AZM; 15 µg), sulphamethoxazole/trimethoprim (SXT; 25 µg), amoxicillin/clavulanate (AMC; 20/10 µg), cefoperazone (CFP; 30 µg), ceftriaxone (CRO; 30 µg), ciprofloxacin (CIP; 5 µg), gentamycin (CN; 10 µg), and fosfomycin (FOS; 50 µg). A fresh colonial suspension (0.5 McFarland bacterial density; 1.5 × 10^8^ CFU/mL) was spread onto a Mueller–Hilton agar (MHA; Oxoid, Cambridge, UK) plate using a sterile swab. The antibiotic discs were fixed onto the medium surface and the plates were incubated at 37 °C for 24 h and then the inhibition zone diameters were recorded. The susceptibility of bacterial isolates to colistin was determined according to the broth microdilution technique as described elsewhere [52]. In brief, colistin was twofold serially diluted in 100 µL Muller–Hinton broth (MHB; Oxoid, Cambridge, UK) in a microtiter plate. An equal amount of MHB with 10^5^ CFU/mL was added. The plates were sealed and then incubated at 37 °C for 24 h. Minimum inhibitory concentrations (MICs) of colistin were analyzed using contemporary breakpoint criteria of colistin at MICs ≤ 2 µg/mL. Strains were defined as extensively drug-resistant (XDR) if they remained susceptible to drugs from at most two classes of antimicrobials, whereas pan drug-resistance (PDR) refers to those resistant to all antimicrobials in all investigated classes [53]. The multiple antibiotic resistance (MAR) indices were evaluated as documented elsewhere [54].

### 4.4. Curing of Plasmids

Plasmid curing of the examined bacterial isolates was performed to attribute the loss of antibiotic resistance markers to the loss of IncF replicons. The examined strains were treated with a curing agent, sodium dodecyl-sulfate (SDS; Sigma-Aldrich, St. Louis and Burlington, MA, USA), as previously described [55]. Briefly, Luria–Bertani (LB; Hi-Media, Mumbai, India) broth containing a subinhibitory concentration of SDS (5%) was inoculated with a 1:10^4^ dilution of freshly isolated colonies and incubated at 44.5 °C for 48 h with constant agitation. Aliquots of the culture were appropriately diluted and then spread onto LB agar plates with and without antibiotics and incubated at 37 °C for 16–18 h. The resulting single clones were streaked onto LB agar medium with and without ampicillin, which was the most prevalent among the antibiotic-resistant isolates. Thus, for the screening convenience of plasmid-loss mutants, ampicillin was used as the priority selective marker. The LB agar plates were incubated at 37 °C overnight, and ampicillin-susceptible clones referred to the loss of the plasmid. The cured strains were subjected to further antibiotic susceptibility profiling using the same panel of antibiotic discs to examine the existence of antibiotic resistance markers.

### 4.5. Plasmid DNA Extraction

Plasmid DNA was extracted and purified from *Salmonella* spp. and *E. coli* isolates before and after curing as previously performed [56]. Briefly, a fresh single colony was inoculated into 2 mL of LB medium and incubated overnight at 37 °C for 24 h, and then bacterial pellets were precipitated at 5000× *g* for 10 min at 4 °C. Plasmid DNA was extracted using the GeneJET Plasmid Miniprep Kit (ThermoFisher Scientific, Waltham, MA, USA) following the manufacturer’s instructions and kept at −80 °C until used.

### 4.6. IncF Replicon Typing

PCR-based IncF replicon typing of FII, FIA, and FIB was accomplished using plasmid extracts of *E. coli* and *Salmonella* isolates before and after curing. PCR amplification was performed by targeting three replicons using specific primer pairs designed for FII, FIA, and FIB (Table 4). All PCR amplifications were performed using the primers with the corresponding published protocol [3,25]. Top Taq Master Mix kit (QIAGEN, Hilden, Germany) was used in compliance with the guidelines provided by the manufacturer. The PCR products were examined after electrophoresis in a 1.5% agarose gel and visualized on a UV transilluminator (Spectroline, New York, NY, USA). The replicons’ products before and after curing were investigated.

### 4.7. IncF Replicon Sequencing and Sequence Analysis

PCR products were purified using a MinElute PCR purification kit (Qiagen, Hilden, Germany) and directly sequenced using an automatic DNA sequencer (Perkin-Elmer, Foster City, CA, USA). FIB amplicons’ sequences of *E. coli* were assigned by submitting the sequences to the pMLST website (www.pubmlst.org/plasmid/, accessed on 20 March 2024). The phylogenetic tree was constructed based on the 80 FIB sequences deposited in the pMLST database using MEGA software (v. 5) [57]. Phylogenetic analysis was performed using the neighbor-joining method [58] and Kimura’s two-parameter model [59] as an estimator for the evolutionary distance among sequences. The topology of the tree was evaluated by 1000 iterations of a bootstrap analysis. Nucleotide analysis of the new allele was performed using the Basic Local Alignment Search Tool (BLAST, https://blast.ncbi.nlm.nih.gov/Blast.cgi/, accessed on 1 April 2024). A new allele (FIB81) was deposited on the pMLST website.

### 4.8. Statistical Analysis

Microsoft Excel (Microsoft Corporation Version 2021) was utilized for data editing purposes. Data were analyzed using the Freq procedure in the Statistical Analytical System, SAS [60], to assess the significant differences in antimicrobial resistance between *E. coli* and *Salmonella* spp. isolates. The statistical significance level was set at *p* < 0.05.

## 5. Conclusions

The study implies that broilers are a significant source of IncF replicons, and the drug-resistant *Enterobacterales* are associated with a highly divergent IncF-FIB plasmid incompatibility group. Tracking of epidemiology and characteristics of resistant CPs is required to generate new efficient treatment policies and infection control strategies.

## Figures and Tables

**Figure 1 ijms-25-08347-f001:**
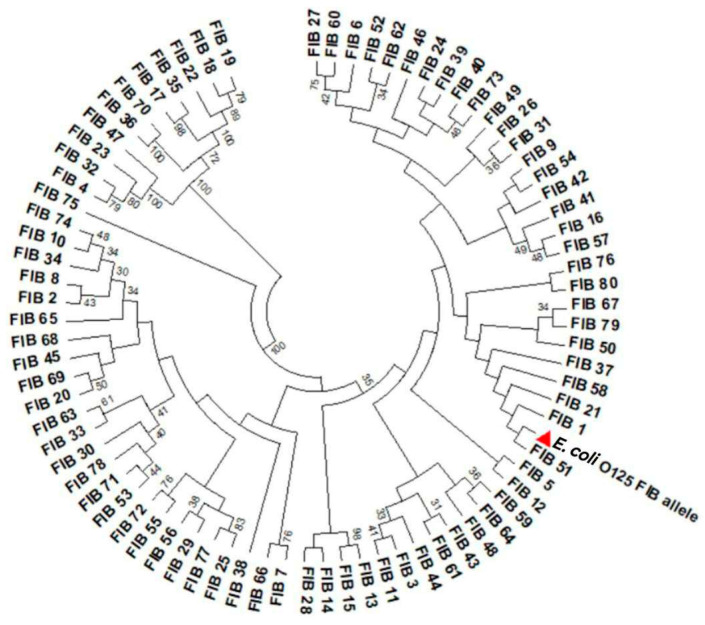
Phylogenetic circle tree of the FIB replicon alleles. The dendrogram was constructed using the neighbor-joining algorithm. Bootstrap values for 1000 replicates are indicated. The distribution of FIB alleles according to point mutations previously revealed eighty alleles. The new allele (FIB81) was examined in *E. coli* (O125) isolates (accession number OR453940) isolated from broiler; it is closely related to the FIB51 allele isolated from a human in France.

**Table 1 ijms-25-08347-t001:** Antimicrobial resistance of *E. coli* (n = 31) and *Salmonella* spp. (n = 36) isolated from broilers.

AMA	No. of Resistant *E. coli* (%)	No. of Resistant *Salmonella* spp. (%)	*p*-Value
AMP	26 (83.87)	31 (86.11)	0.1467
SAM	25 (80.65)	31 (86.11)	0.0889
AMC	20 (64.52)	24 (66.67)	0.3333
CRO	17 (54.84)	14 (38.89)	0.4752
CFP	8 (25.81)	12 (33.33)	0.2925
CTX	18 (58.06)	12 (33.33)	0.1514
DO	15 (48.39)	13 (36.11)	0.6287
CIP	17 (54.84)	9 (25.00)	0.0496 *
CN	14 (45.16)	12 (33.33)	0.6236
C	14 (45.16)	11 (30.56)	0.4577
AZM	18 (58.06)	9 (25)	0.0284 *
SXT	16 (51.61)	13 (36.11)	0.4709
FOS	15 (48.39)	3 (8.33)	0.0010 *
CT	26 (83.87)	31 (86.11)	0.1467

AMP: ampicillin, SAM: ampicillin/sulbactam, AMC: amoxicillin/clavulanate, CRO: ceftriaxone, CFP: cefoperazone, CTX: cefotaxime, DO: doxycycline, CIP: ciprofloxacin, CN: gentamycin, C: chloramphenicol, AZM: azithromycin, SXT: sulphamethoxazole/trimethoprim, FOS: fosfomycin and CT: colistin. * *p*-value < 0.05 was considered statistically significant.

**Table 2 ijms-25-08347-t002:** Antimicrobial susceptibilities of 12 drug-resistant *Salmonella* spp. and *E. coli* isolates.

Strains (n = 12)	Code No.	Sensitivity Pattern	Resistance Pattern
*E. coli* O125 (n = 3)	10	SXT, C	AMP, SAM, AMC, CRO, CFP, CTX, DO, CIP, CN, AZM, FOS, CT
	6	CFP	AMP, SAM, AMC, CRO, CTX, DO, CIP, CN, C, AZM, SXT, FOS, CT
	3	Nil	AMP, SAM, AMC, CRO, CFP, CTX, DO, CIP, CN, C, AZM, SXT, FOS, CT
*S.* Typhimurium (n = 1)	8	SXT, CIP, FOS	AMP, SAM, AMC, CRO, CFP, CTX, DO, CN, C, AZM, CT
*S.* Enteritidis (n = 4)	7	CFP, CRO, FOS	AMP, SAM, AMC, CTX, DO, CIP, CN, C, AZM, SXT, CT
	4	AZM, CN	AMP, SAM, AMC, CRO, CFP, CTX, DO, CIP, C, SXT, FOS, CT
	9	AZM, CN, CFP, FOS	AMP, SAM, AMC, CRO, CTX, DO, CIP, C, SXT, CT
	5	AZM, FOS	AMP, SAM, AMC, CRO, CFP, CTX, DO, CIP, CN, C, SXT, CT
*S.* Pullorum (n = 1)	6	CN, CRO, FOS	AMP, SAM, AMC, CFP, CTX, DO, CIP, C, AZM, SXT, CT
*S.* Gallinarium (n = 2)	10	AZM, CIP, CRO, CFP, FOS	AMP, SAM, AMC, CTX, DO, CN, C, SXT, CT
	1	Nil	AMP, SAM, AMC, CRO, CFP, CTX, DO, CIP, CN, C, AZM, SXT, FOS, CT
*S.* Arizona (n = 1)	2	FOS	AMP, SAM, AMC, CRO, CFP, CTX, DO, CIP, CN, C, AZM, SXT, CT

Resistance patterns include shared resistance of the isolates to ampicillin (AMP), ampicillin/sulbactam (SAM), amoxicillin/clavulanate (AMC), doxycycline (DO), colistin (CT), and cefotaxime (CTX). C: chloramphenicol, AZM: azithromycin, SXT: sulphamethoxazole/trimethoprim, AMC: amoxicillin/clavulanate, CFP: cefoperazone, CRO: ceftriaxone, CIP: ciprofloxacin, CN: gentamycin, CT: colistin, FOS: fosfomycin. Isolate code numbers are underlined in Appendix A.

**Table 3 ijms-25-08347-t003:** Cured antibiotic resistance markers related to IncF-replicons patterns in *Salmonella* spp. and *E. coli*.

Bacterial Isolates	Code No.	Resistance to a Number of Antibiotics/ Cured Resistance Markers	IncF Replicon Pattern before SDS Curing			IncF Replicon after SDS Curing
			FII	FIA	FIB	
*E. coli* O125 (n = 3)	10	12/Nil	−	−	FIB (81)	Nil
	6	13/Nil				
	3	14/C, CN, CFP, CRO, CIP				
*S.* Typhimurium (n = 1)	8	11/AZM, CN	+	+	−	FIIs
*S.* Enteritidis (n = 4)	7	11/CIP	−	+	−	Nil
	4	12/SXT, C, FOS	−	+	+	FIA, FIB
	9	10/CN, CRO	−	−	+	FIB
	5	12/CT C, CFP, CN, SXT, CRO	−	+	−	Nil
*S.* Pullorum (n = 1)	6	11/SXT, C	+	−	+	FIIs, FIB
*S.* Gallinarium (n = 2)	10	9/CN	+	−	+	Nil
	1	14/FOS	−	+	+	FIA, FIB
*S.* Arizona (n = 1)	2	13/Nil	−	+	−	FIA

CRO: ceftriaxone, CIP: ciprofloxacin, CN: gentamycin, CT: colistin, C: chloramphenicol, SXT: sulphamethoxazole/trimethoprim, AZM: azithromycin, CFP: cefoperazone and FOS: Fosfomycin. +: positive, −: negative. Isolate code numbers are underlined in Appendix A.

**Table 4 ijms-25-08347-t004:** Primers used in PCR based replicons typing of the strains.

Bacterial Strain	Primer Name	Primer Sequence (5′–3′)	PCR Run	Amplicon (bP)	References
*Salmonella* spp.	FIA	F: CCATGCTGGTTCTAGAGAAGGTG R: GTATATCCTTACTGGCTTCCGCAG	Duplex	462	[3]
	FIBS	F: TGCTTTTATTCTTAAACTATCCAC R: CTCCCGTCGCTTCAGGGCATT		683	[3,25]
	FIIS	F: CTAAAGAATTTTGATGGCTGGC R: CAGTCACTTCTGCCTGCAC	Simplex	259–260	[25]
*E. coli*	FIA	F: CCATGCTGGTTCTAGAGAAGGTG R: GTATATCCTTACTGGCTTCCGCAG	Duplex	462	[3]
	FIB	F: TCTGTTTATTCTTTTACTGTCCAC R: CTCCCGTCGCTTCAGGGCATT		683	[3,25]
	FII	F: CTGATCGTTTAAGGAATTTT R: CACACCATCCTGCACTTA	Simplex	258–262	[25]

F, forward; R, reverse; bp, base pair.

## Data Availability

The datasets used and/or analyzed during the current study are available from the corresponding author on reasonable request. The FIB nucleotide sequence generated in the current study was deposited into the GenBank under accession number OR453940 (https://www.ncbi.nlm.nih.gov/nuccore/OR453940/, accessed on 1 April 2024). The new FIB81 allele was assigned and published on the pMLST website (https://pubmlst.org/bigsdb?db=pubmlst_plasmid_seqdef&page=alleleInfo&locus=FIB&allele_id=81, accessed on 10 April 2024). The datasets analyzed in the current study are available on request from the corresponding author.

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
