# Peer review of "A Newly Incompatibility F Replicon Allele (FIB81) in Extensively Drug-Resistant Escherichia coli Isolated from Diseased Broilers"

_ijms, 2024, doi:10.3390/ijms25158347_

Round 1

Reviewer 1 Report

Comments and Suggestions for Authors

The authors aimed to characterize E. coli and Salmonella spp. strains isolated from diseased broilers, focusing on the detection of IncF replicons. Even though they present useful results about the dissemination of the replicons and the detection of a novel replicon allele, the manuscript lacks several pieces of information and contains minor/major errors.

Please write the bacteria throughout the text in the correct form (in italics, with the species word starting with a lowercase letter). Please add "spp." to Salmonella where needed throughout the text. The term Enterobacteriaceae has been replaced by the term Enterobacterales.

Why did you select only 12 strains? The number of E. coli isolates tested is limited, although you have isolated 31 strains. Why did you select only three? The antimicrobial susceptibility testing of all isolates should be provided, and the results of the 12 selected strains should be presented separately.

Ln 142 and Table 2: The resistant phenotypes of E. coli strains are not the same. In Table 2, why are some words in bold?

In Table 1: Why do you present only the shared resistance? What about the other five antimicrobials tested? Please add them to the results (highlighting the shared resistance).

How do you explain that the three E. coli strains, even though harboring the same replicon, have different cured resistant markers? Add a comment.

Correct the following errors and change the sentences to be more comprehensive:

Ln 45 (Delete the word “bird”)

Ln49 (Use "mortality" instead of "mortalities.")

Ln52-53, 55-56, 60-62 (use “antibiotic resistant genes through conjugation”), 63-65 (Combine the sentences), 68-72 (write the genes with their respective antibiotics), 88-90, 118 (The term "seriously resistant" is not correct. Clarify what you mean.), 206 (not all strains were XDR), 229-230 (the strains, not the plasmids, can infect humans), 319-320 (Clarify what you mean by "with and without antibiotics." If the sentence relates to lines 320-321, please combine them or rephrase the first sentence).

Comments on the Quality of English Language

The English language usage in the text exhibits several areas needing improvement to enhance clarity and coherence. There are grammatical errors that require attention. Furthermore, scientific notation and terminology must adhere to standard conventions; for instance, bacteria should be italicized, and abbreviations like "spp." in Salmonella should be used.

Author Response

A revision note (Round 1) 

Submission ID: ijms-3094045

Response to Reviewer 1 Comments:

Reviewer 1: The authors aimed to characterize E. coli and Salmonella spp. strains isolated from diseased broilers, focusing on the detection of IncF replicons. Even though they present useful results about the dissemination of the replicons and the detection of a novel replicon allele, the manuscript lacks several pieces of information and contains minor/major errors.

Please write the bacteria throughout the text in the correct form (in italics, with the species word starting with a lowercase letter). Please add "spp." to Salmonella where needed throughout the text. The term Enterobacteriaceae has been replaced by the term Enterobacterales.

Author response: Thanks for the respected reviewer valuable comments. The species of bacteria was written in italic throughout the manuscript. Regarding Salmonella species, the World Health Organization (WHO), Centers for Disease Control and Prevention (CDC) and the American Society for Microbiology (ASM) indicated that, for the species, Salmonella enterica should be italic, while the serovar names should be in Roman type with the first letter capitalized, e.g., Salmonella enterica serovar Typhimurium. After the first use, the serovar may be used without a species name, e.g., Salmonella Typhimurium, which was presented in the revised version of the manuscript. Please find the attached document for Salmonella nomenclature.

Reviewer 1: Why did you select only 12 strains? The number of E. coli isolates tested is limited, although you have isolated 31 strains. Why did you select only three? The antimicrobial susceptibility testing of all isolates should be provided, and the results of the 12 selected strains should be presented separately.

Author response: Thank you for your valuable comments. We selected 12 drug-resistant E. coli and Salmonella spp. isolates, 2 of them were resistant to all tested antimicrobials (PDR) and the other 10 were either XDR (resistant to all tested antimicrobials except 2 or fewer) or MDR with high MAR indices for the IncF replicon typing. The antimicrobial susceptibility testing of all isolates with their statistical analysis was presented in the recently added Table 1 and Supplementary Tables 1 and 2 in the revised version of the manuscript.

Reviewer 1: Ln 142 and Table 2: The resistant phenotypes of E. coli strains are not the same. In Table 2, why are some words in bold?

Author response: Thank you for your insightful comment. The resistance phenotypes of E. coli were revised and corrected in the revised version of the manuscript. The bold words were typing errors, which were corrected in the revised version of the manuscript.

Reviewer 1: In Table 1: Why do you present only the shared resistance? What about the other five antimicrobials tested? Please add them to the results (highlighting the shared resistance).

Author response: Thank you for your insightful comment. Antimicrobial resistance patterns for the 12 selected strains were included in Table 2. Also, we highlighted the shared resistance in the table footnote as per your suggestion.

Reviewer 1: How do you explain that the three E. coli strains, even though harboring the same replicon, have different cured resistant markers? Add a comment.

Author response: Thank you for your comment. As described in Table 3, the cured resistant markers are nil in 2 E. coli strains harbouring the same replicon, which differs from the 3rd E. coli strain. This may be attributed to the complexity of the antibiotic resistance mechanisms of bacteria, which may be chromosomal, or plasmid mediated, particularly in the PDR isolates.

Reviewer 1: Correct the following errors and change the sentences to be more comprehensive:

  • Ln 45 (Delete the word “bird”)
  • Ln49 (Use "mortality" instead of "mortalities.")
  • Ln52-53, 55-56, 60-62 (use “antibiotic resistant genes through conjugation”), 63-65 (Combine the sentences), 68-72 (write the genes with their respective antibiotics), 88-90, 118 (The term "seriously resistant" is not correct. Clarify what you mean.), 206 (not all strains were XDR), 229-230 (the strains, not the plasmids, can infect humans), 319-320 (Clarify what you mean by "with and without antibiotics." If the sentence relates to lines 320-321, please combine them or rephrase the first sentence).

Author response: Thank you for your insightful comments. All corrections were done in the revised version of the manuscript as per your suggestions.

I would like to clarify that plasmid curing is the process of obviating the plasmid-encoded functions such as antibiotic resistance. It converts antibiotic-resistant bacterial cells into sensitive ones. Thus, the elimination of R-plasmids makes the antibiotic therapy effective. As cured derivatives were scored by their failure to grow in the presence of antibiotics, the test was applied with and without antibiotics to confirm the results.

Reviewer 1: The English language usage in the text exhibits several areas needing improvement to enhance clarity and coherence. There are grammatical errors that require attention. Furthermore, scientific notation and terminology must adhere to standard conventions; for instance, bacteria should be italicized, and abbreviations like "spp." in Salmonella should be used.

Author response: Thank you for your comments. We asked a native English-speaking colleague to copy-edit the paper as your recommendation, and he did all the required English editing. Also, the bacteria species was written in italics and the abbreviation spp. was added as per your suggestion.

Reviewer 2 Report

Comments and Suggestions for Authors

A new incompatibility F replicon allele (FIB81) in extremely drug resistant Escherichia coli isolated from diseased broilers

This paper describes screening broilers and the detection of IncF plasmids as well as the indication of a new IncF allele.

Major comments

The abstract is too lengthy and includes some information that can be removed for a more focused paragraph. For example, the full resistance profile is redundant as the most important find is the IncF allele. Also, the fact that the allele was deposited to GenBank and the accession number (line 34-35) are better mentioned in the results section. Please review and edit.

Line 58 – "(http://pubmlst.org/plasmid/, January 30, 2015)" – is this the latest data? It is a very long time ago…

Line 66-68 – "Large conjugative plasmids in a clinical isolate of Klebsiella pneumoniae carried a class 1 integron with blaIMP-4 gene cassette were associated with resistance makers to gentamicin and tobramycin" – this does not contribute to the clarity of the paragraph and would better be removed.

Line 76-78 – a citation to explain were this number has originated (86%) is needed

Line 90-93 – why is this data relevant to your current study? I would remove this, it appears an afterthought at best.

Figure 1 should go to supplementary

The discussion needs editing, as it is not well presented.

The most important part here is, I think, the allele and the suggestion of a program to control its spread. I fear that the paper is not clearly indicating it. You should edit the paper to make the point clearer and more focused.

Conclusions really should not be a summary of the paper but indicate the bottom line

Minor comments

Line 53 – please reword this, as it is it seems disjoined from the rest of the paragraph.

conjugative incompatibility F (IncF) plasmids

Line 65 – what do you mean by "a lot of remarkable encoding resistance"? Please edit for clarity.

Line 140-143 and line 157 – please used italics for species names

Figure 2 – typo - algorithm not algorism

Figure 2 – also - italics for species names please

Line 181-182 – this sentence is disjoined from the rest of the paragraph

Line 231 – you just named the allele – so it cannot be said that it is "known as"

Comments on the Quality of English Language

Some editing is in order - italics are missing and some sentences are disjoined

Author Response

A revision note (Round 1)

Submission ID: ijms-3094045

Response to Reviewer 2 Comments:

Reviewer 2: This paper describes screening broilers and the detection of IncF plasmids as well as the indication of a new IncF allele.

The abstract is too lengthy and includes some information that can be removed for a more focused paragraph. For example, the full resistance profile is redundant as the most important find is the IncF allele. Also, the fact that the allele was deposited to GenBank, and the accession number (line 34-35) are better mentioned in the results section. Please review and edit.

Author response: Thank you for your comment. The abstract was revised and edited. Also, all corrections were done in the revised version of the manuscript as per your suggestions.

Reviewer 2: Line 58 – "(http://pubmlst.org/plasmid/, January 30, 2015)" – is this the latest data? It is a very long time ago.

Author response: Thank you for your valuable comment. This date was deleted from the revised manuscript and the sentence was edited.

Reviewer 2: Line 66-68 – "Large conjugative plasmids in a clinical isolate of Klebsiella pneumoniae carried a class 1 integron with blaIMP-4 gene cassette were associated with resistance makers to gentamicin and tobramycin" – this does not contribute to the clarity of the paragraph and would better be removed.

Author response: Thank you for your comment. This sentence was removed as per your recommendation.

Reviewer 2: Line 76-78 – a citation to explain were this number has originated (86%) is needed

Author response: Thank you for your valuable comment. The reference was added as per your suggestion. Please find the reference below:

Thomas C. M., Nielsen K. M., Mechanisms of, and barriers to, horizontal gene transfer between bacteria. Nat. Rev. Microbiol. 3, 711–721 (2005)

Reviewer 2: Line 90-93 – why is this data relevant to your current study? I would remove this, it appears an afterthought at best.

Author response: Thank you for your comment. This sentence was removed as per your recommendation.

Reviewer 2: Figure 1 should go to supplementary

Author response: Thank you for your comment. Figure 1 was placed in the supplementary materials as per your suggestion.

Reviewer 2: The discussion needs editing, as it is not well presented.

Author response: Thank you for your insightful comment. The discussion was revised, edited and improved as per your suggestion.

Reviewer 2: The most important part here is, I think, the allele and the suggestion of a program to control its spread. I fear that the paper is not clearly indicating it. You should edit the paper to make the point clearer and more focused.

Author response: Thank you for your insightful comment. The manuscript was revised as per your advice.

Reviewer 2: Conclusions really should not be a summary of the paper but indicate the bottom line.

Author response: Thank you for your comment. The conclusion was rewritten to focus on the valuable result of the manuscript.

Reviewer 2:

Minor comments

  • Line 53 – please reword this, as it is it seems disjoined from the rest of the paragraph conjugative incompatibility F (IncF) plasmids

It was revised and edited, thank you.

  • Line 65 – what do you mean by "a lot of remarkable encoding resistance"? Please edit for clarity.

It was edited and clarified, thank you.

  • Line 140-143 and line 157 – please used italics for species names

Thank you for your comment. Regarding Salmonella species, the World Health Organization (WHO), Centers for Disease Control and Prevention (CDC) and the American Society for Microbiology (ASM) indicated that, for the species, Salmonella enterica should be italic, while the serovar names should be in Roman type with the first letter capitalized, e.g., Salmonella enterica serovar Typhimurium. After the first use, the serovar may be used without a species name, e.g., Salmonella Typhimurium, which was presented in the revised version of the manuscript. Please find attached document for Salmonella nomenclature.

  • Figure 2 – typo - algorithm not algorism

It was corrected, thank you.

  • Figure 2 – also - italics for species names please

It was done, thank you.

  • Line 181-182 – this sentence is disjoined from the rest of the paragraph

Thank you for your comment. This statement was removed from the revised manuscript.

  • Line 231 – you just named the allele – so it cannot be said that it is "known as"

Thank you for your comment, it was revised and corrected.

  • Comments on the Quality of English Language: Some editing is in order - italics are missing and some sentences are disjoined.

Thank you for all your efforts to improve the manuscript. I would like to confirm that we asked a native English-speaking colleague to copy-edit the paper as your recommendation, and he did all the required English editing. Also, bacterial species were revised to be italic all over the manuscript.

Round 2

Reviewer 1 Report

Comments and Suggestions for Authors

The authors have made all the suggested changes. The manuscript can be accepted in the current form.